# Avoiding ecosystem and social impacts of hydropower, wind, and solar in Southern Africa's low-carbon electricity system

Grace C. Wu [1,10] ✉, Ranjit Deshmukh [1,2,10] ✉, Anne Trainor[3], Anagha Uppal[4], A. F. M. Kamal Chowdhury [1,5], Carlos Baez[4], Erik Martin[6], Jonathan Higgins[7], Ana Mileva[8] & Kudakwashe Ndhlukula[9]

The scale at which low-carbon electricity will need to be deployed to meet economic growth, electrification, and climate goals in Africa is unprecedented, yet the potential land use and freshwater impacts from this massive build-out of energy infrastructure is poorly understood. In this study, we characterize low-impact onshore wind, solar photovoltaics, and hydropower potential in Southern Africa and identify the cost-optimal mix of electricity generation technologies under different sets of socio-environmental land use and freshwater constraints and carbon targets. We find substantial wind and solar potential after applying land use protections, but about 40% of planned or proposed hydropower projects face socio-environmental conflicts. Applying land and freshwater protections results in more wind, solar, and battery capacity and less hydropower capacity compared to scenarios without protections. While a carbon target favors hydropower, the amount of cost-competitively selected hydropower is at most 45% of planned or proposed hydropower capacity in any scenario—and is only 25% under socio-environmental protections. Achieving both carbon targets and socio-environmental protections results in system cost increases of 3-6%. In the absence of land and freshwater protections, environmental and social impacts from new hydropower development could be significant.

Balancing tradeoffs between energy infrastructure and socio-environmental goals is critical for planning sustainable energy transition pathways. Large hydropower continues to be promoted as a cost-effective and low-carbon source of dispatchable electricity, especially in regions with abundant potential like Africa, Southeast Asia, and Latin America[1–4]. However, the significant negative social and environmental impacts of hydropower projects have historically been underestimated in power sector planning[5–8]. Wind, solar, and battery technologies with their rapidly declining costs have been viewed as promising low-carbon substitutes for new hydropower projects[9–12]. Yet, these technologies have their own technical, environmental, and social challenges. If not addressed, siting conflicts arising from

[1]Environmental Studies, Bren Hall, University of California Santa Barbara, Santa Barbara, CA 93106, USA. [2]Bren School of Environmental Science and Management, University of California, Santa Barbara, CA 93106, USA. [3]Africa Program, The Nature Conservancy, Arlington, VA 22203, USA. [4]Department of Geography, Ellison Hall, University of California, Santa Barbara, Santa Barbara, CA 93106, USA. [5]Earth System Science Interdisciplinary Center, University of Maryland, College Park, USA. [6]Center for Resilient Conservation Science, The Nature Conservancy, Arlington, VA 22203, USA. [7]Global Freshwater Team, The Nature Conservancy, Arlington, VA 22203, USA. [8]Blue Marble Analytics, San Francisco, CA, USA. [9]SADC Centre for Renewable Energy and Energy Efficiency (SACREEE), 11 Dr Agostinho Neto Road, Windhoek, Namibia. [10]These authors contributed equally: Grace C. Wu, Ranjit Deshmukh. ✉e-mail: gracecwu@ucsb.edu; rdeshmukh@ucsb.edu

hydropower, wind, and solar development will likely result in project delays and cost overruns, and require mitigation and compensation costs that could reduce the feasibility of new energy infrastructure critical for achieving universal energy access, energy security, economic growth, and climate goals[5,13,14]. Careful planning of renewable energy infrastructure that accounts for competing conservation needs and societal uses of land is thus critical for rapid and low-impact renewable energy development[15,16].

Much of the literature on the sustainable development of hydropower has focused on minimizing as opposed to avoiding impacts[17–19]. These studies strategically plan a hydropower portfolio by co-optimizing hydropower generation with other socio-environmental criteria[18,20,21], creating optimal hydropower portfolios given a fixed amount of hydropower generation requirements. Isolating hydropower planning from power system planning, which optimally designs the overall technology mix, has benefits in that multiple social and environmental criteria specific to hydropower plants can be considered. However, this approach's main drawback is the inability to identify whether it would be cost-effective to substitute potential higher impact hydropower projects with other generation technologies such as wind, solar, or natural gas[20,22,23]. Studies have recently begun to take a power-systems-level approach to hydropower planning, exploring the potential for wind and solar to replace a certain amount of hydropower capacity. However, these studies substitute hydropower generation on an annual generation basis using solar energy generation potential in nearby locations[24], which overlooks the dispatchable nature of hydropower generation compared to wind and solar. Furthermore, hydropower generation is typically represented as a "fleet" within a power systems planning framework; rarely are individual hydropower projects assessed such that seasonal, daily, hourly, and sub-hourly temporal representation of generation can accurately estimate costs and value of specific hydropower projects. Yet, this is the type of analysis needed to identify which hydropower projects can be cost-effectively substituted[4,25]. While Chowhury et al.[26] and Carlino et al.[27] both examine cost-competitiveness of specific hydropower plants in the African region or subregion in a capacity-expansion framework, finding that about half of proposed hydropower projects are economic, neither attempt to screen projects based on socio-environmental criteria and thus, remaining plants may still impose high environmental or social costs.

While the potential for wind and solar generation is globally abundant, it is variable and uncertain due to weather patterns. Managing this variability will require large battery storage capacities and/or flexible generation like hydropower or natural gas. Thus, as more wind and solar generation substitutes potential hydropower generation, the more valuable hydropower generation becomes, complicating planning and operations of future low-carbon electricity systems and precludes simple one-to-one substitution between generation technologies. Additionally, large-scale wind and solar generation projects, which have significant land use requirements, have also come up against conflicting social, cultural, or environmental uses of land[28,29]. In the United States, more than half of failed renewable energy projects examined were partially or entirely due to environmental impacts, making it the leading cause of project failure[30]. Lake Turkana Wind Farm, Africa's largest wind project, was mired in land tenure controversies involving the improper leasing of community grazing land with cultural significance[31,32]. Failure to consider social and environmental siting criteria in both long-term energy systems planning as well as project-level planning could lead to underestimation of the costs and overestimation of the ease and availability of developing renewable energy infrastructure[5].

The Southern African region epitomizes the conflicts arising from the significant expansion of hydropower and the potential tradeoffs in developing wind and solar projects. The region consists of twelve mainland countries—Angola, Botswana, Democratic Republic of the Congo, Eswatini, Lesotho, Mozambique, Malawi, Namibia, South Africa, Tanzania, Zambia, and Zimbabwe—that together currently account for 40% of Africa's electricity demand, with load projections for 2040 about double the electricity demand in 2022[33]. South Africa alone accounted for 71% of current total electricity consumption in the region in 2021[34]. Eight of these twelve countries, which together comprise the Southern African Power Pool (SAPP), are dependent on hydropower for over half their electricity generation;[35] altogether hydropower accounted for 24% of the overall generation mix in the SAPP in 2021[36]. Southern Africa is home to two of the five largest river basins in Africa—the Zambezi and Congo—with tens of gigawatts of proposed hydropower projects. With declining costs of wind and solar photovoltaics (PV), the region also has the opportunity to scale up its renewable energy generation[26]. At the same time, the region has large areas with high biodiversity value[37]. Protecting these areas and avoiding social conflicts with local communities will be critical for Southern Africa to sustainably develop its wind, solar, and hydropower resources.

In this study, we characterize wind, solar, and hydropower potential in the SAPP and identify the mix of electricity generation technologies that would be cost-minimizing under different sets of socio-environmental constraints and carbon emissions targets. We ask, how does excluding the most socially and environmentally damaging potential wind, solar, and hydropower projects impact optimal electricity pathways and overall system costs in Southern Africa? To create socio-environmentally constrained scenarios, we screen wind, solar, and hydropower techno-economic potential using protected areas, sensitive habitat for focal species, areas of importance for cultural values and livelihoods, forested areas, and free-flowing rivers. We then supply these screened candidate projects to a power system capacity investment model of the SAPP, GridPath-SAPP, to create cost-optimal electricity generation portfolios and identify the wind, solar, and hydropower projects that would remain cost-competitive under each scenario[26]. To explore the implications of reaching social, conservation, and climate objectives concurrently, we compare scenarios that do and do not cap annual carbon dioxide emissions. For the carbon cap target, we use a linear trajectory to achieve 50% of 2020 emissions by 2040 for the entire region. This target is roughly consistent with South Africa's 2030 Nationally Determined Contributions (NDCs), or post-2020 climate actions in accordance with the Paris Agreement. Other countries in the region have a much wider range of commitments (e.g., 15% below 2010 levels by 2030 for Botswana or 91% below business as usual for Namibia) or currently lack 2030 NDCs.

## Results
### Environmentally and socially-constrained wind, solar, and hydropower projects
Wind and solar resource potential was quantified using a combination of technical, physical, economic, and socio-environmental criteria (Supplementary Table 3). For hydropower, using available project-specific design specifications, we modeled the reservoir footprint of 34 major planned or proposed projects that met data quality and size specifications (comprising about 20 gigawatts or GW of nameplate capacity out of a total of about 41 GW of planned/proposed capacity; the remaining 21 GW were included in the model). To design electricity portfolios that avoid negative environmental and social impacts of new development, we constrained the techno-economic wind and solar resource potential and planned/proposed hydropower projects by excluding areas with varying levels of environmental and/or social importance to produce the following seven screened scenarios of candidate renewable energy resources (the scenarios are named using italics; see Methods and Supplementary Tables 1–3 for detailed scenario definitions): (1) *Base*, in which all planned or proposed hydropower projects are included and no socio-environmental exclusions are applied for hydropower; legal exclusions are applied to wind and

solar, (2) *Legal*, in which strictly legally protected areas (e.g. International Union for Conservation of Nature (IUCN) I-II) are excluded from wind, solar, and hydropower potential, (3) *Social*, in which legally protected and areas important for human livelihoods as well as planned hydropower plants whose reservoirs would displace more than 5000 people are excluded, (4) *Environmental*, in which other non-strictly protected areas (e.g. IUCN III-VI) that allow for other activities and high conservation value areas as well as planned hydropower projects on or impacting large free flowing rivers are excluded, (5) *Environmental and Landscape*, in which *Legal*, *Environmental*, and forested areas are excluded, (6) *All Exclusions*, in which all *Legal, Social, Environmental and Landscape* areas are excluded, and (7) *All Exclusions No New Hydropower*, in which *All Exclusions* have been applied for wind and solar development and no new hydropower can be developed.

In response to all socio-environmental land protections (*All Exclusions*), we find that solar potential decreases substantially, with only 17% of the *Base* scenario potential remaining, whereas wind potential decreases less dramatically with about 72% of the *Base* potential remaining in the *All Exclusions* scenario (Fig. 1). The technical potential for wind power is generally far more limited than solar power

even under the base scenario (4.5 TW of wind vs. 20 TW of solar) and thus, any siting protections or land use exclusions will naturally reduce solar potential more than wind potential (Fig. 1d, e). Landscape exclusions account for a significant reduction in solar potential (Fig. 1d). While nearly all countries still have large amounts of solar potential that is more than sufficient to meet all domestic electricity demand on an annual basis in the *All Exclusions* scenario, most of the remaining potential is concentrated in South Africa, Namibia, Botswana, and Angola (Fig. 1a). Wind potential is also widely distributed across countries even with socio-environmental protections, although in Angola, Mozambique, and the Democratic Republic of Congo (DRC) wind potential is limited to smaller areas (Fig. 1b) due to either low wind speeds and/or extensive forest cover. Planned hydropower capacity reduces to 58% (about 25 GW) of total planned potential in the *All Exclusions* scenario. About 12% of planned hydropower project capacity overlaps with legally protected areas, and a further 17% overlaps with areas of high conservation value or is sited on a large free-flowing river. In response to the combined exclusions for land, freshwater, and community protections, all countries experience a significant reduction (>50%) in planned hydropower capacity except for DRC (Fig. 1c).

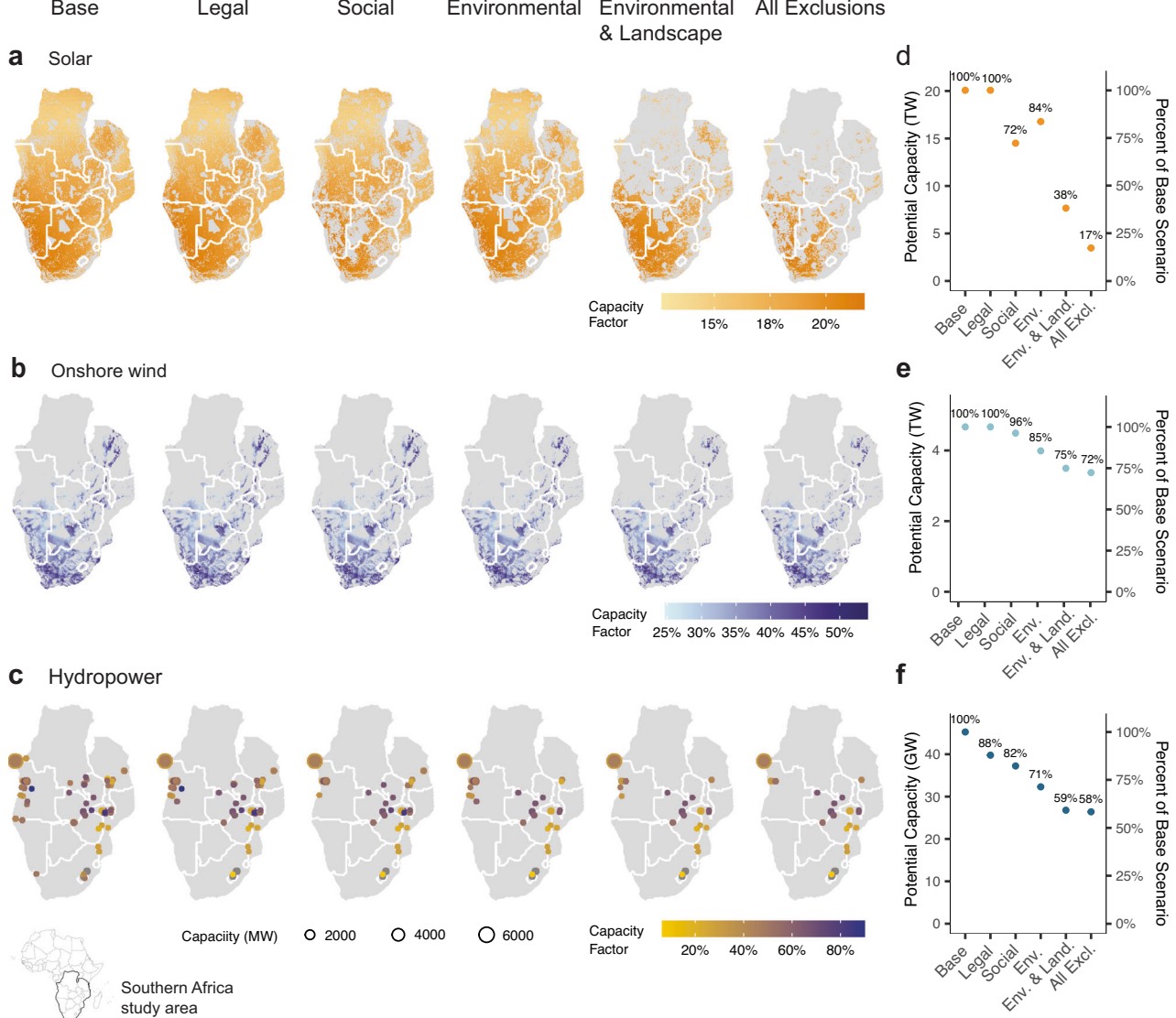

**Fig. 1 | Solar photovoltaic, onshore wind, and hydropower resource potential in Southern Africa under various socio-environmental scenarios.** Distribution of renewable resource potential spatial (**a**–**c**), and total capacity and shares compared to base scenario (**d**–**f**) for solar photovoltaic, onshore wind, and hydropower by scenario. *Base* and *Legal* scenarios are the same for solar and onshore wind.

## Optimal electricity portfolios

In order to understand how imposing siting exclusions shifts future optimal generation capacity requirements, electricity system costs, emissions, and socio-environmental impacts, we used an open source capacity expansion model, GridPath, to develop least-cost electricity pathways for Southern Africa from 2020 to 2040[26]. We assume an electricity demand forecast from the base scenario in the Southern African Power Pool Plan[33]. For scenarios that lack a carbon target, we capped all scenarios' carbon emissions to the *Base* scenario's annual emissions in each investment period, thereby holding carbon emissions the same across the no-carbon-target scenarios.

We find that without socio-environmental siting exclusions, the Southern African region will need 176 GW of new cost-optimal generation capacity installed by 2040, which is about 130% increase over existing capacity in 2020. Wind and solar technologies account for about half of this new capacity. This drives the share of wind and solar generation from 4% to 55%. Increasing environmental and social protections across all renewable technologies requires investing in more solar, battery, and gas, while building less hydropower, with the differences due to siting protections resulting in a change of less than 10% of the total new capacity in the *Base* scenario (*Legal* to *All Exclusions* scenarios in Fig. 2a). A hydropower moratorium, on the other hand, results in the most significant deviations from the *Base* scenario—requiring a substantial increase in wind, solar, battery, and gas capacity (*All Exclusions No New Hydropower* scenario in Fig. 2a) and generation (Supplementary Fig. 5a). Importantly, increasing siting protections, including the hydropower moratorium, results in no new coal capacity and even decreases in generation from existing coal power plants. Siting protections reduce selected wind capacity while increasing solar capacity because high quality wind sites were excluded by socio-environmental constraints. Overall capacity difference trends for the no-carbon target scenarios do not monotonically change with increasing siting protections—that is, about 8 GW of additional gas capacity is required in the *Environmental* and *Environmental + Landscape* scenarios, whereas only 1.5 GW of new gas is required in the *All Exclusions* scenarios (Fig. 2a).

A low-carbon target alters the way that socio-environmental protections impact the energy portfolio. To achieve a 50% reduction in carbon emissions by 2040 without any additional socio-environmental protections in place (*Base*), about 50 GW of additional wind, solar, and hydropower capacity (compared to the *Base* scenario without a carbon target) will be required (Fig. 2b) to fill the gap left by reduced fossil fuel generation (Supplementary Fig. 5b). Achieving a high level of socio-environmental protections in addition to a low-carbon target requires a growth of largely wind (+7.5%), solar (+29%), and battery (+23%) capacity, and a reduction of hydropower capacity (−31%; Fig. 2b), resulting in a net gain in capacity of 43% by 2040 in the *All Exclusions* scenario compared to the *Base* without carbon target scenario. Legal protections with a low-carbon target have the greatest single impact on total capacity increases (+4.2%), followed by Landscape protections (an additional +3.4%). Overall, under a low-carbon target, no additional gas capacity is selected as more socio-environmental protections are imposed (except under a hydropower moratorium) despite more gas generation along with battery storage being utilized to balance the additional solar.

Hourly dispatch profiles for representative days highlight the important roles that wind and solar play in meeting electricity demand during the low hydropower generation months (dry season; July–October)—suggesting high seasonal complementarity between wind and hydropower—and the role that hydropower plants play on a diurnal basis by ramping up generation during the evening hours (Supplementary Fig. 3). While only 1.5 GW of new gas capacity is added in the *All Exclusions* scenario (2% increase) under no-carbon targets, gas generation increases by 23% to compensate for the lack of load-following generation resulting from increases in solar and reductions in hydro and wind capacity (Fig. 2a, Supplementary Fig. 5a).

Unlike many other linear capacity expansion models which treat candidate resources as fleets of generation as opposed to individual projects[38], we developed the capacity investment model as a mixed integer program to identify whether or not it would be cost effective to build each planned or proposed hydropower plant. We find in the *Base* scenario, only about 18 GW and 13 GW of hydropower will be needed in 2040 with and without a carbon target, respectively (Fig. 2). Given that 41 GW of planned or proposed hydropower (not including pumped hydro) capacity was available, we find that a large fraction of these candidate hydropower projects are never cost competitive. Applying socio-environmental protections further reduces this selected capacity by 5.5 GW with a carbon target and by 3 GW without, such that only 25% of planned or proposed hydropower capacity (approx 10–12 GW) is necessary and cost-competitive by 2040.

The geographic distribution of selected (determined by GridPath-SAPP to be cost-effective), suitable but not selected (suitable in that scenario, but not determined to be cost-effective), and unsuitable hydropower projects (screened out in a scenario) shows that some

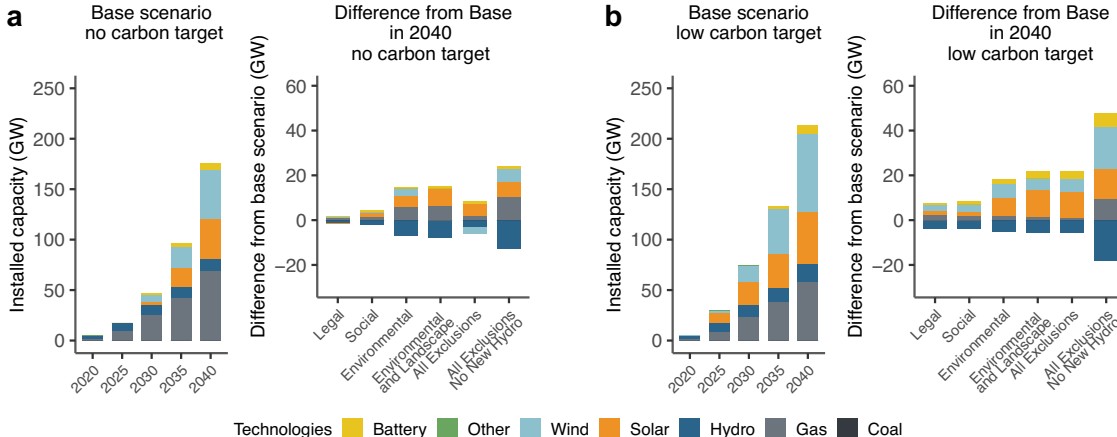

**Fig. 2 | New capacity in gigawatts (GW) for modeled generation and storage technologies in 2040 for various socio-environmental scenarios. a** New generation capacity installations from 2020 to 2040 for the *Base* scenario without a carbon target and differences in installed capacities in 2040 for each scenario compared to *Base*. **b** Same as (**a**) but with a low carbon emissions target trajectory that limits annual carbon emissions in 2040 to half of carbon emissions in 2020. Positive differences indicate greater installed capacity and negative differences indicate lower installed capacities compared to the *Base* scenarios. Source data are provided as a Source Data file.

cost-competitive proposed projects have potential negative social and/or environmental impacts, which exclude them from development in the more protective scenarios (Fig. 3). These hydropower projects are primarily located in Angola, Mozambique, Tanzania, and Zambia, with more than half of the proposed projects in Mozambique, Zambia, and Tanzania excluded due to environmental impacts and most of the projects in Angola excluded due to landscape impacts (Fig. 3d). The most cost-competitive proposed hydropower projects are concentrated in the Kwanza and Zambezi river basins due to their high annual productivity (i.e., capacity factors of >50% on average). Some large hydropower projects in the Congo river basin/DRC region (Inga 3 with ~13 GW of proposed capacity) are also suitable across all scenarios, but these relatively expensive projects are selected only after the projects in the Kwanza, Zambezi, and Rufiji river basins have already been selected and/or excluded due to socio-environmental protections (in the *All Exclusions* scenario). The most significant differences in hydropower build-out between a low-carbon target and no-carbon target are observed in Angola where several more GW of hydropower capacity is cost competitive under a low-carbon target (Supplementary Fig. 7e).

These differences in geographic distribution of hydropower projects between scenarios resulted in only modest differences in additional international transmission capacity requirements (Supplementary Fig. 6). There is slightly more transmission capacity on the corridor joining Tanzania, Zambia, and Namibia and slightly less transmission capacity between Botswana and South Africa with more

siting protections and/or less hydropower selected. This was likely due to more proportional compensating increases in wind, solar PV, and/or natural gas capacity additions domestically.

## Carbon emissions, systems costs, and avoided impacts of portfolios

Carbon emissions without the low-carbon cap slightly increase from 2020 to 2035 while sharply decreasing by more than 10% in 2040, largely due to coal retirements and a drop in the costs of wind and solar PV technologies (Fig. 4a). For the low-carbon scenarios, we capped emissions at levels that linearly decrease with investment periods to meet a carbon target that achieves 50% reduction of 2020 emissions by 2040.

Marginal electricity system costs increase in response to more socio-environmental protections. However, despite the significant amount of high impact potential excluded from the model for onshore wind, solar, and hydropower, system costs only increase 0.4% due to *Legal* exclusions, 1.8% due to *Environmental* exclusions, and up to 2.3% due to *All Exclusions* (Fig. 4b). These cost premiums for socio-environmental protections do increase further when combined with a low-carbon target, which avoids 100 million tonnes of annual $CO_2$ emissions in 2040 or approximately 350 million tons of $CO_2$ cumulatively between 2020 and 2040 (Fig. 4a). A 3% cost increase is required under the *Base* scenario to achieve the low-carbon target, a premium that increases to 4% in the *All Exclusions* scenario. Pursuing both a low-carbon target and all socio-environmental protections increases costs by 6.8% compared to the *Base* scenario with no carbon target.

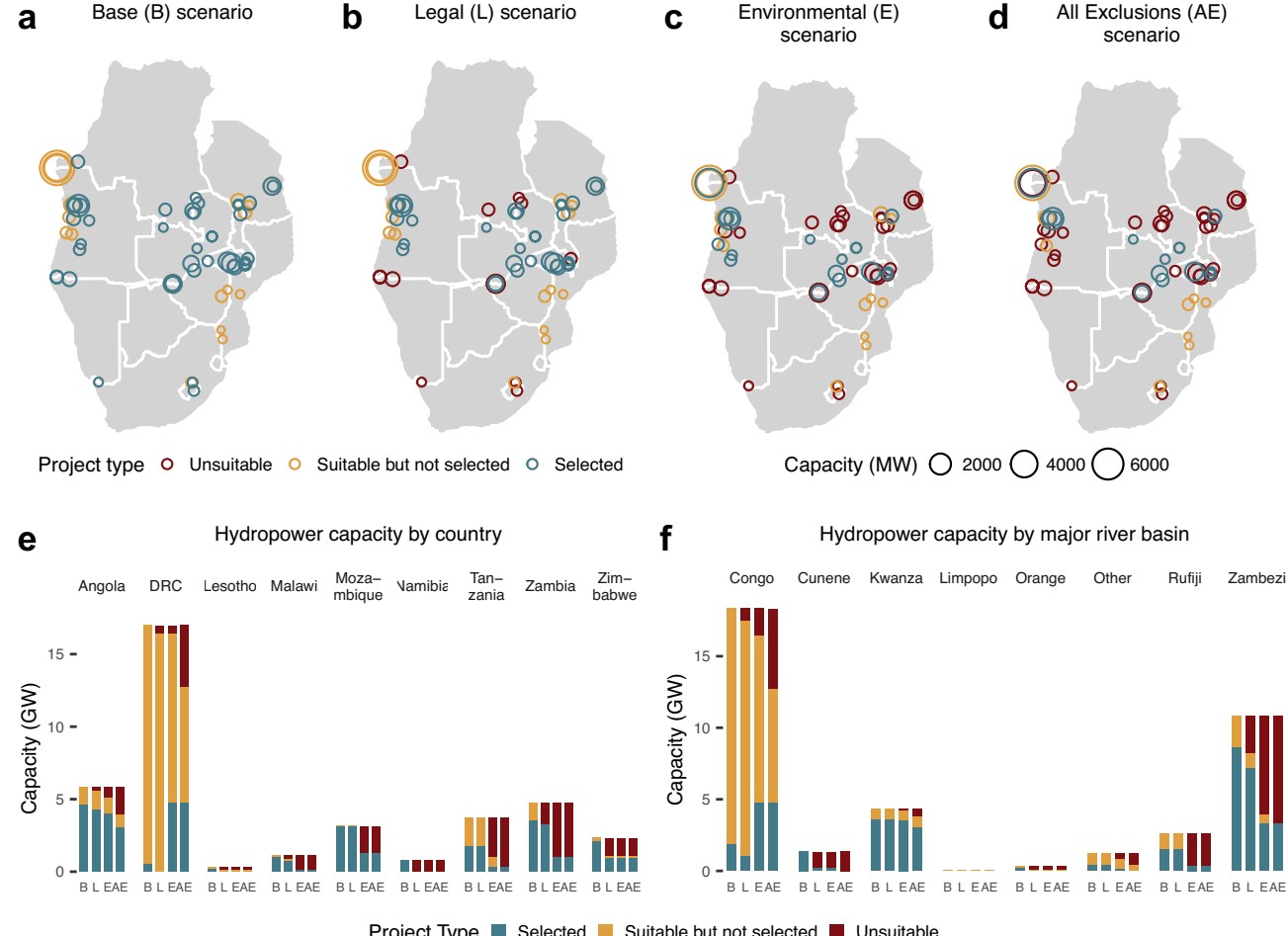

**Fig. 3 | Hydropower capacity of select socio-environmental scenarios in the low-carbon target case.** Spatial distribution in megawatts (MW) (**a**–**d**), country-wise capacities in gigawatts (GW) (**e**), and basin-wise capacities (**f**) of unsuitable, suitable but not selected, and selected hydropower projects for *Base* (B), *Legal* (L), *Environmental* (E), and *All Exclusion* (AE) scenarios for the low-carbon target case. Source data are provided as a Source Data file.

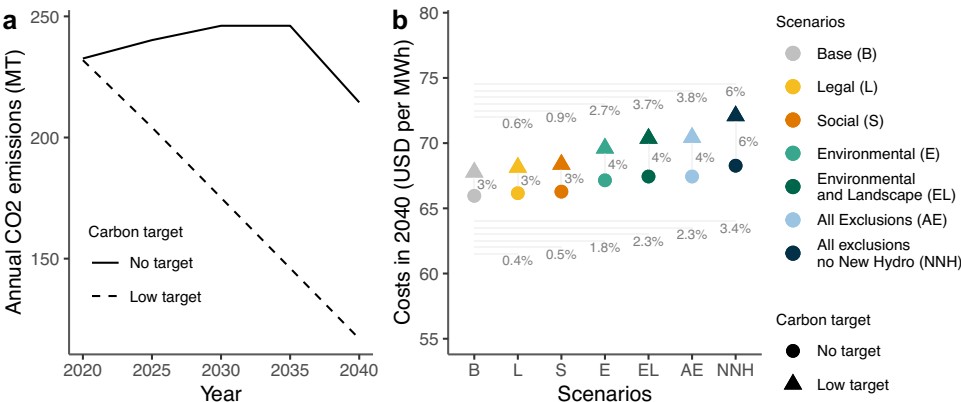

**Fig. 4 | Annual carbon dioxide (CO₂) emissions and electricity systems costs in 2040. a** Carbon dioxide emissions expressed in megatonnes (MT) and (**b**) 2040 costs of Southern Africa's electricity system for the no-carbon target and low-carbon target scenarios expressed in US dollars (USD) per megawatt-hour (MWh). Carbon emissions for all scenarios are either capped at the *Base* scenario without an emissions reduction target or the *Base* scenario with a low carbon emissions target trajectory. Carbon emissions in the low-carbon scenarios are capped at levels that linearly decline over time to meet a 50% emissions reduction target by 2040

compared to 2020. Cumulative emissions between 2020 and 2040 are 28% lower in the low-carbon target scenarios (3500 vs. 4850 MT CO₂). Annual emissions in 2040 are nearly 50% lower in the low-carbon target scenarios. Horizontal lines and corresponding percentages at the ends of each line show the relative cost increase between each scenario and the *Base* scenario (**b**). Vertical lines and the corresponding percentages show the cost increases between the no-carbon target and the low-carbon target scenarios (**b**). Source data are provided as a Source Data file.

We quantified the degree to which social and environmental land and freshwater protections effectively avoid loss of natural habitat and agricultural lands due to hydropower development. About 400 sq km of legally protected areas and over 1500 sq km of conservation areas could have been inundated under the no-protections *Base* scenario in which all proposed projects were supplied to the capacity investment model (Fig. 5). Under the Environmental scenarios with a low-carbon target, all impacts assessed were reduced by more than half, including the number of people displaced. It is notable that in the *Base* scenario, more than 150,000 people could be displaced. Applying social screens on projects reduces the number of people displaced to less than 20,000 (Fig. 5). Avoided human population and cropland impacts for selected hydropower projects are similar for scenarios with no carbon target, but are lower for conservation areas and rangelands in the base scenario with no carbon target (Supplementary Fig. 8).

## Discussion

Using spatially detailed models representing hydrological, solar, and wind generation potential in combination with a power systems investment and operations model, we demonstrate that it is possible to avoid wind, solar, and hydropower development in areas with the greatest social and terrestrial and freshwater ecosystem impacts with relatively modest increases in system costs.

A large fraction of proposed hydropower project capacity has high environmental and social impacts. In contrast, there is plentiful low-impact solar and wind potential. Proposed hydropower projects in Southern Africa have the potential to severely alter many remaining free-flowing rivers in sensitive and biodiverse river basins[19,39] as well as displace hundreds of thousands of people, largely in minority and Indigenous communities[40]. Through a systematic overlay of planned/ proposed hydropower projects with protected and conservation areas, population density, croplands, and rangelands in the Southern Africa region, we find that about 40% of planned/proposed hydropower capacity has significant social and/or environmental impacts. A sizable fraction—12%—of proposed/planned hydropower project capacity will negatively impact legally protected areas through inundation or significant alterations in the degree of regulation of rivers running through legally protected areas. These results largely agree with accounts from media and studies of the socio-environmental impacts of individual dams such as Steigler's Gorge[41], Batoka Gorge[42], Epupa and Baynes[43,44], and Kholombidzo[45].

Although candidate areas for solar PV projects also decline significantly due to land use protections, the remaining potential is more than sufficient for meeting forecasted demand. While candidate areas for wind projects have less geographic overlap with socio-environmental protections, leaving large wind resource potential, avoided areas tend to have high wind resource quality (and thus lower cost), resulting in an overall reduction in the amount of wind in the lower impact portfolios in the absence of a carbon target. These results −abundant high quality solar, lower high quality wind after considering social and environmental land use exclusions−are consistent with studies in other geographies[16,46−48]. Wind and solar's relative abundance, even after applying social and environmental protections, ensure that they are cost-saving substitutes for a large majority of planned and proposed hydropower projects in Southern Africa.

Avoiding or minimizing negative land and freshwater impacts results in notable shifts in the relative investments of wind, solar, hydropower, and natural gas, but does not result in additional coal capacity. In the absence of a carbon target, solar PV and natural gas capacity are the most cost competitive technologies and offset avoided high-impact hydropower capacity as well as avoided new coal capacity (even in the absence of an emissions targets). However, we find that with a carbon target, low-impact wind and solar development coupled with battery storage become the dominant substitutes for high-impact hydropower. Imposing a carbon cap increases the cost-competitiveness of hydropower as a source of low-carbon electricity in higher impact scenarios, but when social and environmental exclusions are applied, the reliance on hydropower is very similar to unconstrained emissions scenarios−only 25% of planned or proposed hydropower capacity or 10 GW is necessary and cost-competitive by 2040. This is generally consistent with other studies that explore power system portfolios with varying shares of hydropower capacity and found that solar PV and wind capacity coupled with flexible gas generation are the primary substitutes for hydropower[9,11,12,27]. Studies that have examined optimal technology investments in response to restricting wind and solar PV development due to environmental protections found an overall shift from wind to solar PV[16,47], which we observe for some, but not all, scenarios in Southern Africa. A comparison with basin-level hydropower results from Carlino et al. (2023) reveals some notable differences in how much hydropower is deemed cost competitive in Southern Africa[27]. Carlino et al. (2023) found that 12−13 GW of hydropower capacity was selected in the Congo basin

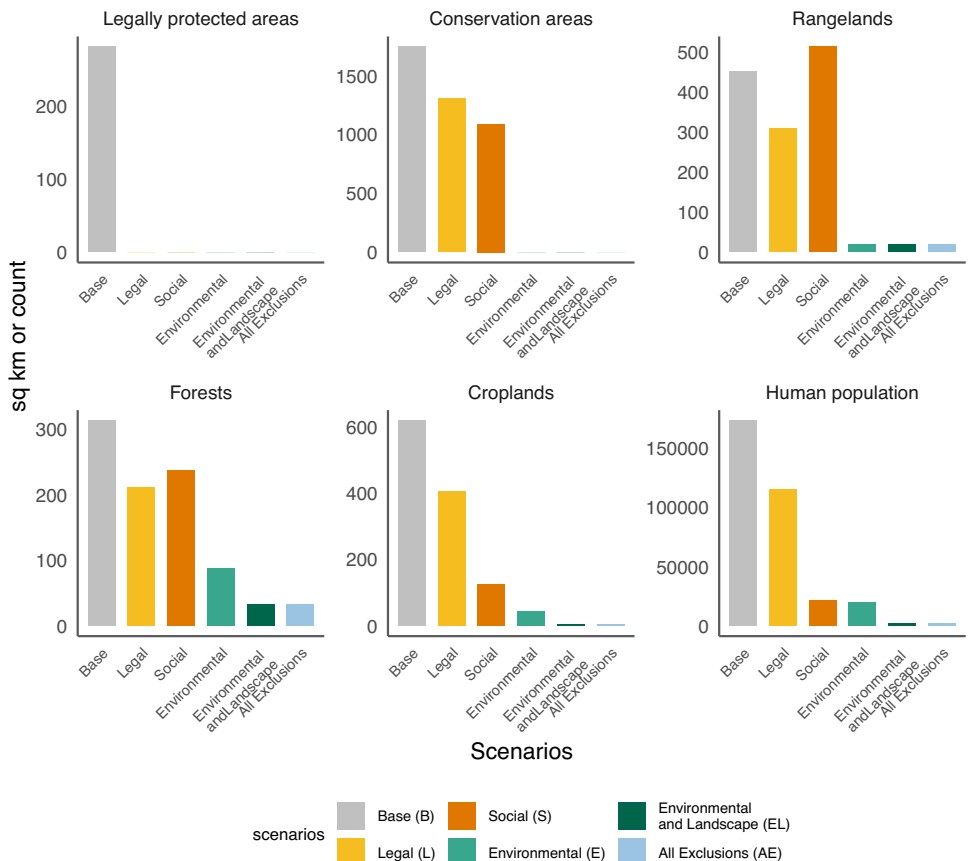

**Fig. 5 | Environmental and social impacts of hydropower projects selected in the scenarios in the low-carbon target case.** Environmental and social impacts (in square kilometers (sq km) of inundated area or number of people displaced) of selected hydropower projects for scenarios meeting the low-carbon target. Source data are provided as a Source Data file.

across all scenarios, whereas our results found only 2–4.8 GW of Congo basin projects to be cost-competitive. Results for the Zambezi basin are more aligned (3–4 GW when considering Environmental or All Exclusions, which is consistent with the higher climate impact scenarios in Carlino et al.), but the composition of specific hydropower projects differ between the two studies.

System costs increase modestly with social and environmental protections, but more notably when including all social, environmental and climate objectives. Achieving an environmentally sustainable, more socially equitable, and low-carbon electricity system for Southern Africa does incur higher costs compared to unrestricted, higher-impact development−3.8% more for *All Exclusions* and 6.8% more for *All Exclusions* with a low-carbon target. While a moratorium on new hydropower development is technically feasible, it does result in the highest, though still modest, cost increases of 3.4% above *Base* without a carbon target and 6% above *Base* with a carbon target. Intermediate impact scenarios result in more modest cost increases (0.4–3.7%). These cost premiums are largely aligned with other studies that have examined the cost of restricting hydropower development[12] or developing lower impact utility-scale wind and solar projects while meeting climate targets[47]. It is also critical to note that the cost and project timeline overruns due to higher impact energy projects are well documented and not accounted for in this study; 80% of large hydropower projects experience cost overruns, with the mean cost overrun being 96%[5]. While we did not quantify the cost differences for each country, we expect that costs will vary across countries depending upon which hydropower and renewable energy projects are excluded from consideration. Improving electricity trade and transmission infrastructure could mitigate costs and impacts on

consumers[26]. Given the region's low contribution to historical carbon emissions and considering its development needs, the international community should consider supporting the additional costs of environmentally and socially sustainable low-carbon pathways.

In the absence of land and freshwater protections, environmental and social impacts from new hydropower development could be significant. We find that several proposed hydropower projects with high environmental and social impacts are cost-competitive (e.g., have high capacity factors, low capital costs) and thus are selected in cost-optimal portfolios when hydropower development is unconstrained. Notably, these high impact scenarios result in several hundred or thousands of square kilometers of biodiverse areas directly inundated and potentially more indirectly impacted (e.g., roads, transmission lines, substations) by the presence of a dam. The construction of these cost-competitive hydropower projects in the *Base* scenarios could also displace upwards of 150,000 people. The vast majority of these impacts can be avoided in the low impact scenarios. We note, however, that while we did quantify degree of regulation and alterations of free-flowing rivers, we did not quantify other environmental impacts of dams on rivers typically considered in strategic dam planning studies− e.g., sediment transport, GHG emissions from inundation, fish diversity, and river fragmentation[11,17]−due in part to data limitations and the fact that some impacts are portfolio-dependent (e.g., sediment transport depends on other dams in the water basin). Future studies that more fully integrate the essential elements of both strategic dam planning and power system planning could better enable sustainable and more holistic low-carbon energy planning.

While many energy infrastructure projects result in negative social and environmental impacts, this study shows that it is possible

to avoid the most damaging impacts for the dominant low-carbon electricity generation technologies—wind, solar PV, and hydropower—while meeting a 50% reduction carbon target and the anticipated growth in electricity demand in the Southern Africa region.

## Methods

The analysis was conducted in four major stages, as illustrated in Fig. 6. Below, we detail the steps in each stage as well as describe the development of the environmental and social scenarios.

### Environmental and social scenarios

We developed environmental and social screens for candidate and planned wind, solar, and hydropower projects (Supplementary Tables 1 and 2) using publicly available and government agency acquired data (Supplementary Table 3). The *Base* scenario for wind and solar screens is the same as the *Legal* scenario. We included all planned hydropower plants in the *Base* scenario even if they overlapped with legally protected areas.

For the *Legal* (Legally Protected) scenario, we excluded renewable potential in areas identified as International Union for Conservation of Nature (IUCN) categories "Ia" (strict nature preserves), "Ib" (wilderness areas), "II" (national parks) in the World Database on Protected Areas (WDPA). We used only categories I and II in this scenario as these areas typically have the strictest level of government protection. For example, category Ib are wilderness areas and category II are National Parks. There are no category Ia areas within the study region. Since some countries do not follow IUCN categories, this scenario also excluded country-specific "National Park" designations in South Africa DRC, Lesotho, Angola, Eswatini, and Tanzania.

For the *Environmental* scenario, we excluded areas identified as IUCN categories "III" (natural monument or feature), "IV" (habitat or species management area), "V" (protected landscape or seascape), "VI" (protected area with sustainable use of natural resources). We also excluded 27 more refined designations that focus on conservation and management of natural resources (e.g., Ramsar sites, which are important wetland sites identified under the international environmental treaty of the Ramsar Convention, reserves, and game management areas). Similar to legally protected areas, country-specific

data on conservation and management areas from DRC, Tanzania, Angola, Zambia, Malawi, Namibia and South Africa were excluded in this scenario. We also excluded Key Biodiversity areas to account for important areas for species conservation. In the *Environment and Landscape* scenario, we account for intact forest by excluding areas with dense forest cover defined as areas with greater than 15% tree cover (based on ESA CCI 2018 dataset—categories 50, 60, 61, 62, 70, 80, 90, 160, and 170) and areas designated as wetlands (based on Global Lakes and Wetlands Database, WWF). See the hydropower characterization subsection below for how these datasets were used to screen hydropower projects.

For the *Social* (Human Livelihoods) scenario, to protect lands with social or cultural value, we excluded areas from the WDPA that focused on human livelihood benefits (e.g., world heritage sites, catchments, community conservancies and reserves). Similar to legally protected areas, we excluded country-specific data on human livelihood benefits (e.g., community forests in Namibia). Due to the strong reliance on subsistence agriculture in the region, irrigated and rainfed croplands were also excluded in the social impact.

Lastly, we developed a sixth scenario, *All Exclusions*, containing both *Environmental and Landscape* and *Social* scenario exclusions, as well as a seventh scenario, *All Exclusions No New Hydropower*, with the All Exclusions screens for wind and solar but a moratorium on new hydropower projects (i.e., no new hydropower).

### Wind and solar resource mapping

We adapted and built upon the Multi-criteria Analysis for Planning Renewable Energy (MapRE) modeling framework, which was first developed for and applied to regions in Africa[49] and recently applied specifically to Southern Africa[26] in addition to regions of the US[16]. MapRE is a spatial renewable energy potential modeling framework that integrates renewable resource assessment and estimation of multiple criteria for decision making analysis. Using spatially explicit wind and solar average resource data sets[50,51], constraints on elevation and slope[52,53], and the environmental and social screens, we identified candidate wind and solar PV sites for the seven environmental and social impact scenarios across the twelve Southern African countries. We conducted the site-suitability analysis at a spatial resolution of

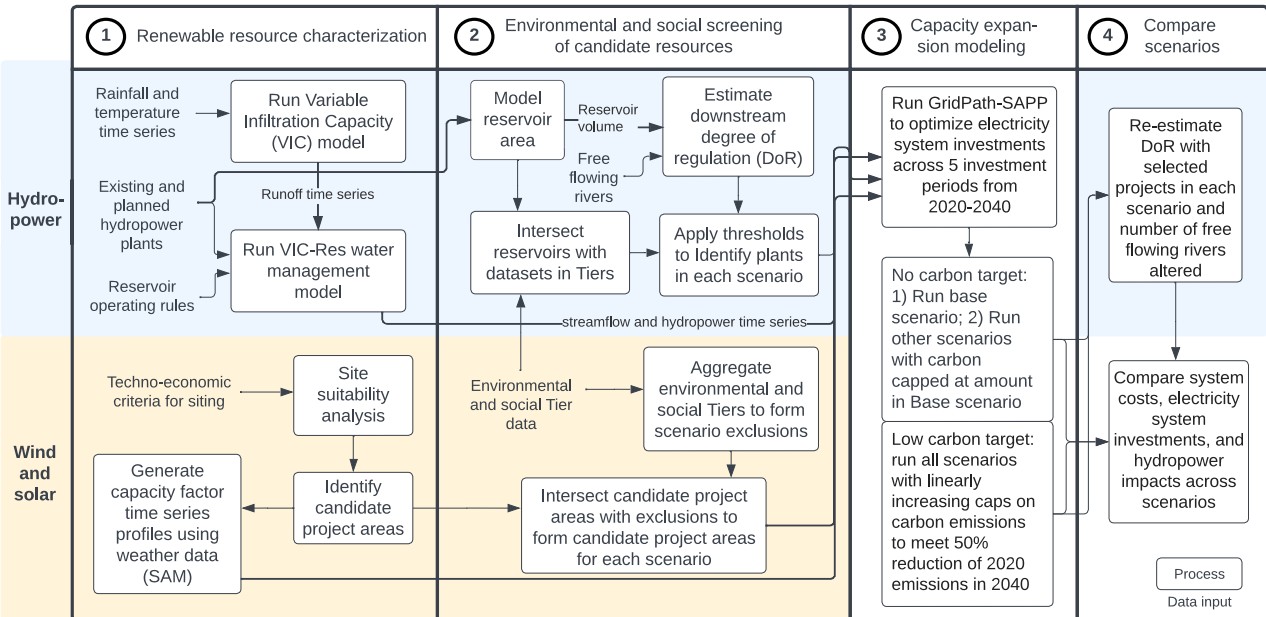

**Fig. 6 | Methods workflow.** The methodology is divided into four distinct stages—(1) renewable resource characterization, (2) environmental and social screening of candidate resources, (3) capacity expansion modeling, and (4) comparison of scenarios. Blue colored areas indicate methods specific to hydropower and light orange colored areas indicate methods specific to wind and solar photovoltaic technologies.

500 m, and then aggregated sites to create candidate project areas that have a maximum size of 25 sq km and 100 sq km for solar PV and wind, respectively, which roughly corresponds to 1 GW of solar PV and 300 MW of wind installed capacity, respectively.

Next, we developed hourly capacity factor time series for the centroids of each wind and solar PV candidate project area using 2018 weather data. For wind, we used hourly wind speed data from ERA5 (European Center for Medium-Range Weather Forecasts - ECMWF - Reanalysis 5)[54], applying a linear bias correction to the coarse spatial resolution data to match the annual average wind speeds from the finer spatial resolution Global Wind Atlas (GWA) data, following the approach detailed in Chowhury et al.[26]. This approach is comparable to Sterl et al.[55]. We then applied a Vestas 2 MW 90 m turbine power curve to the modified hourly wind speeds to derive hourly capacity factors using the System Advisor Model[56]. For solar PV, we used hourly global horizontal irradiance (GHI) data from the National Solar Radiation Database (NSRDB) derived from the Meteosat satellite[57]. We again used the System Advisor model[56] to convert GHI data to capacity factors for fixed tilt systems, setting the tilt equal to the latitude of each location. The economics of each wind and solar candidate project area is determined largely by the resource quality (capacity factor), the spur line costs, and road costs. We estimated costs of transmission interconnections (spur lines) and roads using distance of candidate projects to the nearest transmission and road infrastructure and then applied capital costs assuming a new 230 kV High Voltage Alternating Current (HVAC) transmission line[58] and an asphalt road. For each environmental and social scenario, the list of wind and solar candidate projects and their capacities and costs were then provided as inputs to the power systems planning model, GridPath.

## Hydropower characterization and screening

We first mapped existing and planned hydropower projects using latitude and longitudes of project sites primarily from the SAPP master plan[33], supplemented by Zarfl et al.[59]. We then generated energy availability data for each existing and planned hydropower project using a spatially-distributed hydrological water management model, described below. We modeled eight river basins—Zambezi, Congo, Kwanza, Cunene, Rufiji, Orange, Limpopo, and Buzi—which encompass more than 90% of SAPP's total installed (13 GW) and planned/proposed (41 GW) hydropower capacity[33]. There are 90 proposed and planned projects (status is 'candidate' or 'committed') with the following characteristics:

- 3 GW (25 dams) are not located in the 8 major river basins and were thus designed as in the "Other basin", which included 1.7 GW of projects that lacked locations.
- 789 MW of projects in the "Other basin" has capacity less than 100 MW (14 dams)
- 1.1 GW (20 dams) were deemed small and thus their reservoirs were not modeled. The majority of these (14 or 789 MW) lacked locations and/or were located in an "Other basin".
- 2.5 GW (6 dams) were extensions of existing projects
- 38 GW (54 dams) have socio-env screens
- 20 GW (37 dams) reservoir storage volume and extent were modeled and specific impacts were assessed, but about 16.4 GW (or 13 dams) share the same screens as other dams due to proximity on the same stretch of the river.

To simulate daily runoff, evaporation, and baseflow, we first used the Variable Infiltration Capacity (VIC) hydrological model for each basin[60]. The gridded runoff simulated by VIC was then routed through the river network by VIC-Res, a water management model that simulates daily river discharge as well as the storage and release dynamics of each hydropower project's reservoir[61]. The water release for each reservoir was determined by dam-specific rule curves accounting for the reservoir water level, inflow, storage capacity, and downstream water requirements (for irrigation and other purposes). Using these

average daily energy budgets estimated by VIC-Res, GridPath-SAPP determines the hourly dispatch for each reservoir within a maximum limit (determined by the rated capacity) and a minimum limit (assumed to be 30% of the average daily energy budget), ensuring maximum generation for each day does not exceed the daily energy budget. The design specifications of existing and planned reservoirs were retrieved from global reservoir and dam databases[59,62], and complemented by basin-specific studies on Zambezi[63], Congo[64], Cunene[65], Kwanza[66], Rufiji[67], and Orange[68]. For more details on the hydropower power potential characterization methodology and validation, see[26].

To model the reservoir storage areas for each existing and planned/proposed hydropower project point location, we used the project locations, dam and hydraulic head heights, and a 90 m Digital Elevation Model (DEM)[69]. For each basin, we first filled the DEM to eliminate sinks (internal drainages) and generated flow direction and flow accumulation rasters, using ESRI's ArcMap 10.7. We then manually modified dam locations to ensure spatial alignment between the dam point and flow path, as represented in the flow accumulation raster. We then estimated the reservoir surface elevation by adding the hydraulic head height to the elevation at the dam point taken from the DEM. Next, the contributing watershed for each dam location was estimated, using the "watershed" tool in the ArcMap 10.7 Spatial Analyst extension, with the flow direction raster and dam location as inputs. A reservoir surface raster was then derived by setting all elevation values less than the reservoir surface elevation, within the extent of the contributing watershed, to the reservoir surface elevation. The reservoir surface raster and the filled DEM were then used to calculate the reservoir volume using the difference in elevation between the filled DEM and the reservoir surface elevation raster. This reservoir modeling process was scripted in Python primarily using ESRI's arcpy package. We then validated our estimated reservoir areas against reported existing and potential areas in the literature or through internet search and modified the hydraulic head heights of particular dams when necessary to ensure approximate alignment with the literature. We compared the modeled reservoirs of existing dams to satellite imagery of existing reservoirs to validate the overall reservoir modeling process. See Supplementary Fig. 1 for a map showing all modeled reservoirs.

To determine the suitability of each proposed hydropower project under each environmental and social scenario, we estimated the area of each criteria (e.g., protected areas, environmental datasets, community forests, rangelands, forested areas, croplands) within modeled reservoirs area that could be inundated. See Supplementary Fig. 2 for examples of screened hydropower projects and their reservoirs overlaid on top of environmentally and culturally sensitive areas. For the *Legal* scenario, we excluded reservoirs that inundated any areas with IUCN level I & II protection and sites designated as national parks. We also excluded dams that would cause a significant degree of regulation (>=20) of rivers running within or bordering IUCN level I and II areas. The degree of regulation (DOR) is a measure of potential flow alteration from dams that is calculated for each stream segment and is based on the cumulative storage volumes in upstream reservoirs. A DOR = 1 indicates that 1% of a river segment's mean annual flow could be stored in upstream reservoirs. Although DOR does not capture actual flow alteration resulting from specific dam operations, particularly at sub-annual time scales, it is a useful indicator of alteration potential that can be generated without extensive hydrologic and operational data. Six of the 20 dams excluded in the *Legal* scenario were due to DOR impacts in protected areas. For the remaining scenarios, we applied percentage inundated thresholds for excluding specific projects. For each respective scenario, we excluded dams whose modeled reservoirs' overlap with the aggregated environmental datasets exceeded 5% of the reservoir area (*Environmental* scenario); modeled reservoirs have forest land use efficiency less than 500 MWh/acre of forest land inundated (*Environmental and Landscape*

scenario) based on the land use efficiency of solar in the US;[70] modeled reservoirs' overlap with world heritage sites, community forests, forest reserves (or other important natural or cultural resources that have livelihood benefits) exceed 5% of total reservoir area or would displace more than 5000 people (*Social* scenario).

For the *Environmental* scenario, we also excluded projects based on whether they would alter a free-flowing river. Using Grill et al.[71], we identified free flowing rivers (Connectivity Status Index = 1) and classified them into two groups based on river size class (determined by long-term average discharge using logarithmic progression). Small rivers were defined as rivers with average discharge <100 m3/s (river order > 4) and large rivers with average discharge >100 m3/s (river order ≤4). We then excluded proposed projects located directly on large free flowing rivers. To account for the potential downstream impact of dams on free-flowing rivers, we also calculated the potential DOR for each stream segment downstream of a dam[62]. We excluded projects that were expected to significantly alter large free-flowing rivers downstream (DOR > 100). Downstream segments with DORs exceeding 100 indicate upstream multi-year reservoirs that "have the ability to release water in accordance with an artificial, demand driven regime, often with the explicit goal to supply water in contrast to natural expectations, such as by increasing dry-season flows or eliminating flood peaks" (Lehner et al 2011). The screened lists of hydropower projects for each scenario were then provided to the power system planning model as candidate projects. See Supplementary Table 4 for summary of the outcomes of the hydropower project screening process, including technical characteristics of each hydropower plant.

## Power system planning and impacts

To identify cost-optimal electricity infrastructure investments in the SAPP for each of the scenarios, we used GridPath-SAPP model[26], built on the GridPath open-source power system modeling platform[72]. Utilizing temporal and spatially-explicit demand, wind, solar, and hydro resource data along with various economic and technical constraints, GridPath's capacity-expansion functionality identifies cost-effective deployment of conventional and renewable generators, storage, and transmission lines by co-optimizing power system operations and infrastructure investments.

The GridPath-SAPP model has 12 load zones, each representing a SAPP member country. These load zones are joined by transmission corridors that have existing, planned, and candidate transmission capacities. We modeled five investment periods—2020, 2025, 2030, 2035, and 2040—each representing 5 years. To account for end effects (costs incurred beyond the model planning time horizon), we also included 2045 as an investment period representing 10 years. The model was allowed to choose to build new infrastructure or retire existing infrastructure during an investment period. We assumed a common 7% discount rate for each investment period to calculate the net present value of costs incurred during that period.

Within each investment period, grid infrastructure is dispatched to meet load and other constraints over 24 h during 12 days, each representing a month, and weighted appropriately to represent a full year. Energy demand and supply is balanced in each modeled hour for each load zone. Hydropower and battery storage energy availability is constrained over each day.

The model co-optimizes investments (over each 5-year period) in new system infrastructure including generation, storage, and transmission, and hourly operating costs, while meeting country-wise hourly electricity demand, technical constraints on generators, storage, and transmission lines, and other policy constraints (e.g., emissions reduction targets). The model assumes perfect foresight for electricity demand and technology and fuel costs. New generation capacities are selected linearly except for hydropower projects, which are discretely selected (binary decision). Annual capacity build rates for all technologies except for hydropower (which have specific start years) are not

constrained. GridPath is written in Python and uses the Pyomo optimization language[73]. The Gurobi solver was used for all simulations[74].

Key inputs to GridPath include projected hourly electricity demand for each investment period, installed and candidate generation capacities, hourly capacity factors of wind and solar generators, monthly energy availability of hydropower projects, and existing capacities and unit investment costs of transmission infrastructure. Hourly time series of electricity demand are based on actual 2018 data linearly extrapolated across investment periods assuming growth rates from the base scenario of the SAPP Plan[33]. This scenario assumes a doubling of electricity demand from 2020 to 2040 and assumes an average electricity consumption of 1600 kWh per capita by 2040. Electricity demand is assumed to be inelastic and does not respond to changes in electricity costs. Existing generation capacities—mostly composed of hydropower, coal, and natural gas, with small shares of nuclear, oil, diesel, biomass, wind and solar PV—are adopted from the SAPP Plan[33]. Installed coal plants are assumed to retire at an age of 55 years.

Candidate coal and gas plants are available only in countries with existing capacities of those technologies. Candidate wind and solar capacities and discrete hydro power plants vary based on the environmental and social scenarios described earlier. Wind, solar, and battery storage costs are from the SAPP Plan[33] and their trajectories are adopted from the National Renewable Energy Laboratory's Annual Technology Baseline projections[75]. Only mid-case trajectories are considered in this study. Coal and natural gas fuel cost projections are from the SAPP Plan. Emission factors for fuels are from the Energy Information Administration[76].

Other techno-economic parameters of the generators including fixed operating and maintenance (O&M) costs, variable O&M costs, heat rates, fuel costs, start-up costs, ramp rates, minimum operating levels, minimum up and down times, capital costs, plant lifetimes, emission per unit generation, storage charging and discharging efficiencies, and transmission losses are adopted from the SAPP Plan[33], South Africa's Integrated Resource Plan[77], and[26]. Primary reserve margin (PRM) of 15% over peak demand is imposed as a constraint for new capacity investments. Only dispatchable generation and storage technologies and only 10% of wind capacity can contribute to PRM.

We assumed full coordination among the SAPP countries, with only transmission losses and transfer capacities as constraints to electricity trade. Existing interconnection transfer capacities are adopted from the SAPP[78,79]. GridPath optimally builds new transmission capacities along existing and planned transmission corridors. Lengths of the interconnectors are estimated using the centroids of countries. Investment costs for new transmission lines and substations are from the Western Electricity Coordinating Council[58]. We assume bulk transmission losses of 1% per 100 miles[80].

Major outputs are new-built capacities of generation, storage, and transmission, hourly electricity dispatch, curtailment, and transmission losses, exports and imports among the countries, operating and investment costs, and $CO_2$ emissions.

## Data availability

Source data are provided with this paper. All data supporting the findings of this study are available in the Supplementary Information or are available in Figshare with the accession code: https://doi.org/10.6084/m9.figshare.24492043. Sources for raw data inputs are available in Supplementary Table 3 and generally publicly available unless noted otherwise. Data needed to run GridPath-SAPP have been deposited on Zenodo under accession code: https://zenodo.org/records/10067530.

## Code availability

Code to run the analysis in this study are available at the following links. MapRE: https://github.com/cetlab-ucsb/mapre; VIC-Res: https://github.com/Critical-Infrastructure-Systems-Lab/VICRes; GridPath: https://github.com/blue-marble/gridpath.

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

## Acknowledgements

This project was funded with UK Aid from the UK government under the Applied Research Programme on Energy and Economic Growth (EEG), managed by Oxford Policy Management. The authors thank the Southern African Power Pool Coordination Center and its member utilities for providing data. The study used computational facilities purchased with funds from the National Science Foundation (CNS-1725797) and administered by the Center for Scientific Computing (CSC). The CSC is supported by the California NanoSystems Institute and the Materials Research Science and Engineering Center (MRSEC; NSF DMR 2308708) at UC Santa Barbara.

## Author contributions

GCW, RD, AT, JH conceptualized the study. RD, AT, AFMKC, GCW, AU developed the methodology. AFMKC, RD, AM, GCW designed the software. AT, AU, AFMKC, RD, GCW, EM conducted the formal analysis. AT, AU, AFMKC gathered and curated the data. GCW led the drafting of the original manuscript. GCW, RD, AT, AU, AFMKC, EM, JH, AM, KN reviewed, contributed to, and edited the manuscript. RD, GCW, KN, AT solicited and received the funding. GCW, CB produced the visualizations. GCW, RD, AT provided supervision.

## Competing interests

The authors declare no competing interests.
