## [Peer review file · Nature Communications]

Avoiding ecosystem and social impacts of hydropower, wind, and solar in Southern Africa's low-carbon electricity systemREVIEWER COMMENTS

Reviewer #1 (Remarks to the Author):

The paper characterises wind, solar, and hydropower potential in the Southern African Power Pool and identifies the mix of electricity generation technologies that would be cost-minimizing under different sets of socio-environmental constraints and carbon emissions targets. The work has a great potential to contribute to the energy planning of the Southern African Region, in the energy and related fields. While the work is well written and the methodology is sound, there is a need to restructure the sections so that the results are analysed after the presentation of the methodology. Also, the discussion is to be followed by a conclusion at the end.

Other minor edits are:

I. Line 20 – “...technologies (e.g., wind, solar, natural gas) (20,22,23).” Remove the first bracket - could read “... technologies such as wind, solar or natural gas (20,22,23).”

II. Line 10 -11 “... with projections for 2040 double that of current demand (31).” - something is missing in this sentence

III. Line 6- “... capacity increases (+4.2%), followed by Landscape protections (an additional 3.4 percentage points)” – if plus means additional percentage points then for consistency 3.4 should be +3.4%; and “(2% increase)”= +2%

IV. Line 38 - (Fig 3D)- use d for consistency

V. Line 3 – “... Indigenous...” lower case

VI. Line 14 – Authors to capture somewhere in the paper how river protections impact PV projects

Reviewer #3 (Remarks to the Author):

This is a well-written and well thought out paper focusing on including environmental protection criteria in capacity expansion models for the Southern African Power Pool.

I enjoyed reading the paper and found it easy to read and understand. Moreover, it covers an important subject that deserves to be published. Before I can fully recommend the paper for publication, I would like the authors to consider below comments:

P2 Line 25-30: “rarely are individual hydropower projects assessed” – I would recommend to cite and discuss here a recent publication in Science (Carlino et al. 2023) which did exactly this, <https://www.science.org/doi/10.1126/science.adf5848>. It may be worth comparing the results and/or the main conclusions of the present paper with Carlino et al. in the discussion later, too.

P2 Line 32: “Large battery storage capacities or flexible generation like (...)” – I would say “and/or”

P2 Line 10: “that together currently account for 40% of Africa’s electricity demand” – I would mention here how much of this is from South Africa alone, too; and the share of hydropower in the region’s electricity generation.

P4 Fig 1: I would explain that the reason why wind potential reduces less than solar potential under the given restrictions, is that wind potential is anyway less spatially distributed than solar potential, and so it is less affected by e.g. the forestry criterion, since wind is anyway too weak in the Congo rainforest to exploit commercially. Basically, the reduction for solar PV potential comes across as dramatic, but in reality it does not really matter, as solar PV can be built almost anywhere and the remaining potential more than covers potential future needs anyway.

P4 Fig 1: P5 Section 2.2: It should be made clear here which demand growth is assumed over the modelling period, and to which per-capita electricity consumption by 2050 this corresponds.

P8 Fig 3: I would also show the buildout of transmission lines under the different scenarios in this figure. Do higher environmental constraints for hydro and VRE mean more or less need for cross-regional exchanges?

P14 Line 38: “adjusting the coarse spatial resolution data to match the annual average wind speeds from the finer spatial resolution Global Wind Atlas (GWA) data”—can the authors expand on this? Is it similar to the method described in Sterl et al. 2022 <https://www.nature.com/articles/s41597-022-01786-5>? (In fact, I would generally recommend to have a look at that paper which assesses model-ready datasets for solar PV and wind power across Africa, also taking into account land use constraints and protected areas: <https://www.nature.com/articles/s41597-022-01786-5>.)

P15 Line 26: “The water release for each reservoir was determined by dam-specific rule curves accounting for the reservoir water level, inflow, storage capacity, and downstream water requirements”—If this is the case, then how did the capacity expansion model decide on the hydropower dispatch at sub-daily level? It sounds as if the release was pre-ordained before model entry, but I am not sure if this was really the case.

Reviewer comments and author responses for # NCOMMS-23-32341A

“Avoiding ecosystem and social impacts of hydropower, wind, and solar in Southern Africa’s low-carbon electricity system”

Grace C. Wu, Ranjit Deshmukh, Anne Trainor, Anagha Uppal, AFM Kamal Chowdhury, Carlos Baez, Erik Martin, Jonathan Higgins, Ana Mileva, Kudakwashe Ndhlukula

- In this document, **actions taken to revise the manuscript are in bold typeface**; explanations of the revisions are in non-bold typeface.
- See the revised manuscript with track changes.

Reviewer #1

ID	REFEREE COMMENT	AUTHORS’ RESPONSE AND EXPLANATION OF REVISIONS
R1-1	Key suggestions are underlined The paper characterises wind, solar, and hydropower potential in the Southern African Power Pool and identifies the mix of electricity generation technologies that would be cost-minimizing under different sets of socio-environmental constraints and carbon emissions targets. The work has a great potential to contribute to the energy planning of the Southern African Region, in the energy and related fields. While the work is well written and the methodology is sound, there is a need to restructure the sections so that the results are analysed after the presentation of the methodology. Also, the discussion is to be followed by a conclusion at the end.	Actions taken to revise the manuscript are in bold; explanations of these revisions (where appropriate) and response to the comment follow in non-bold formatting. We thank the reviewer for the supportive comments and suggestions that helped improve the readability of the manuscript. As per Nature Communications Guide to authors, for the “Article” content type (https://www.nature.com/ncomms/submit/article), the organization of the sections is as follows: Introduction, Results, Discussion, Methods. See below for the excerpt from the Guide: The main text of an Article should begin with a section headed Introduction of referenced text that expands on the background of the work (some overlap with the abstract is acceptable), followed by sections headed Results, Discussion (if appropriate) and Methods (if appropriate). The Results and Methods sections should be divided by topical subheadings; the Discussion should be succinct and may not contain subheadings. As such, the structure of our original submission adheres to these guidelines, with the presentation of the Methods following the Discussion and we do not include a Conclusion section as this is not allowed. However, at the end of the introduction, we provide an overview of our approach and at the beginning of the Results section, we further elaborate on the scenario design and model assumptions so that readers can interpret most Results without reading the Methods section.
R1-2	Other minor edits are: I. Line 20 – “...technologies (e.g., wind, solar, natural gas) (20,22,23).”	Thank you for this suggestion. We’ve made the recommended minor changes.

	Remove the first bracket - could read "... technologies such as wind, solar or natural gas (20,22,23)."	
R1-3	II. Line 10 -11 "... with projections for 2040 double that of current demand (31)." - something is missing in this sentence	Thank you for flagging this. We've rephrased this sentence to read (changed in green): "with load projections for 2040 about double the electricity demand in 2022 (31)."
R1-4	III. Line 6- "... capacity increases (+4.2%), followed by Landscape protections (an additional 3.4 percentage points)" – if plus means additional percentage points then for consistency 3.4 should be +3.4%; and "(2% increase)"= +2%	Thank you for flagging this inconsistency in use of percentage point increase vs. percentage increase. We've changed this sentence to read (change in green): "Legal protections with a low-carbon target have the greatest single impact on total capacity increases (+4.2%), followed by Landscape protections (an additional +3.4%)."
R1-5	IV. Line 38 - (Fig 3D)- use d for consistency	Thank you for catching this inconsistency. We've changed this to "Fig. 3d."
R1-6	V. Line 3 – "... Indigenous..." lower case	We have chosen to capitalize Indigenous as per the definition adopted by the United Nations, as explained in this SAPEIENS article (https://www.sapiens.org/language/capitalize-indigenous/). Capitalized Indigenous refers to "people of long settlement and connection to specific lands who have been adversely affected by incursions by industrial economies, displacement, and settlement of their traditional territories by others" as opposed to the Webster dictionary definition of lower case indigenous, which means, "existing, growing, or produced naturally in a region or country; native".
R1-7	VI. Line 14 – Authors to capture somewhere in the paper how river protections impact PV projects	We are not certain which specific page the reviewer is referring to, but we deduce based on sequencing of comments that the reviewer is referring to the below sentence, "Although candidate areas for solar PV projects also decline significantly due to land use and river protections, the remaining potential is more than sufficient for meeting forecasted demand" in the Discussion section. We apologize that this is an error on our part and thank you for flagging it. We did not intend to include "river protections" for solar PV candidate project areas. As per Table S2 (Scenarios) copied and pasted below, large free flowing rivers and degree of regulation were not applied to solar PV areas. We have changed the sentence to read, "Although candidate areas for solar PV projects also decline significantly due to land use protections, the remaining potential is more than sufficient for meeting forecasted demand."

Table S2. Scenarios

Scenario name	Tiers excluded	
	Wind and solar	Hydropower
Base	Tier A	None (all planned and proposed projects)
Legal	Tier A	Tier A
Social	Tier A + B	Tier A + B
Environmental	Tier A + C	Tier A + C + large free flowing rivers
Environmental and Landscape	Tier A + C + D	Tier A + C + large free flowing rivers + D
All Exclusions	Tier A + B + C + D	Tier A + B + C + D + large free flowing rivers
All Exclusions No New Hydro	Tier A + B + C + D	All planned/proposed hydropower excluded

Reviewer #3

ID	REFEREE COMMENT	AUTHORS' RESPONSE AND EXPLANATION OF REVISIONS
Reviewer #3		
R3-1	This is a well-written and well thought out paper focusing on including environmental protection criteria in capacity expansion models for the Southern African Power Pool. I enjoyed reading the paper and found it easy to read and understand. Moreover, it covers an important subject that deserves to be published. Before I can fully recommend the paper for publication, I would like the authors to consider below comments:	We thank the reviewer for these supportive and constructive comments that helped improve clarity and contributions of the manuscript. We hope the below responses and revisions address any outstanding issues.
R3-2	P2 Line 25-30: “rarely are individual hydropower projects assessed” – I would recommend to cite and discuss here a recent publication in Science (Carlino et al. 2023) which did exactly this, https://www.science.org/doi/10.1126/science.adf5848. It may be worth comparing the results and/or the main conclusions of the present paper with Carlino et al. in the discussion later, too.	Thank you for pointing us to this newly published paper. Since Carlino et al. (2023) was published while our manuscript was under review, we could not compare our approach or results with theirs in our initial submission. However, we agree with the reviewer that this newly published paper is highly relevant and similar to our study. We briefly note the similarities and differences in our additional text in the Introduction, below, as well as compare results with additional text in the Discussion, below. We have added the following sentence to the Introduction: “While Chowhury et al. (2022; (26)) and Carlino et al. (2023; (27)) both examine cost-competitiveness of specific hydropower plants in the African region or subregion in a capacity-expansion framework, finding that about half of proposed hydropower projects are economic, neither attempt to screen projects based on socio-environmental criteria and thus, remaining plants may still impose high environmental or social costs.” We have added the following sentence to the Discussion: “A comparison with basin-level hydropower results from Carlino et al. (2023) reveals some notable differences in how much hydropower is deemed cost competitive in Southern Africa (27). Carlino et al. (2023)

		found that 12 - 13 GW of hydropower capacity was selected in the Congo Basin across all scenarios, whereas our results found only 2 - 4.8 GW of Congo Basin projects to be cost-competitive. Results for the Zambezi Basin are more aligned (3-4 GW when considering Environmental or All Exclusions, which is consistent with the higher climate impact scenarios in Carlino et al.), but the composition of specific projects differ between the two studies.” We note here other differences, but do not include them in the main text of the manuscript as Carlino et al.’s results on additional new capacity are summarized for the entire continent. Carlino et al. builds at least 200 GW of new coal capacity by 2050, whereas our scenarios do not build any new coal, even under the no-carbon cap cases. Also, new capacity additions are dominated by solar in Carlino et al., whereas we see fairly equal additions of wind, solar PV, and natural gas capacity, with more wind than solar or natural gas in the low carbon target case.
R3–3	P2 Line 32: “Large battery storage capacities or flexible generation like (...)” – I would say “and/or”	Thank you for this suggestion. We have implemented this change in the manuscript.
R3–4	P2 Line 10: “that together currently account for 40% of Africa’s electricity demand” – I would mention here how much of this is from South Africa alone, too; and the share of hydropower in the region’s electricity generation.	Thank you for this suggestion. We’ve added the following (in green) to the manuscript: “that together currently account for 40% of Africa’s electricity demand, with load projections for 2040 about double the electricity demand in 2022 (31). South Africa alone accounted for 71% of total electricity consumption in the region in 2021 (32). Eight of these twelve countries, which together comprise the Southern African Power Pool (SAPP), are dependent on hydropower for over half their electricity generation (33); altogether hydropower accounted for 24% of the overall generation mix in the SAPP in 2021 (34).”
R3–5	P4 Fig 1: I would explain that the reason why wind potential reduces less than solar potential under the given restrictions, is that wind potential is anyway less spatially distributed than solar potential, and so it is less affected by e.g. the forestry criterion, since wind is anyway too weak in the Congo rainforest to exploit commercially. Basically, the reduction for solar PV potential comes across as	Thank you for this suggestion to increase the clarity of the explanation of land use protections on resource potential. As suggested we have made the following additions to the Results section (in green): “The technical potential for wind power is generally far more limited than solar power even under the base scenario (4.5 TW of wind vs. 20 TW of solar) and thus, any siting protections or land use exclusions will naturally reduce solar potential more than wind potential (Fig. 1d-e). Landscape exclusions account for a significant reduction in solar potential (Fig 1d). While nearly all countries still have large amounts of solar potential that is more than sufficient to meet all

	dramatic, but in reality it does not really matter, as solar PV can be built almost anywhere and the remaining potential more than covers potential future needs anyway.	domestic electricity demand on an annual basis in the All Exclusions scenario, most of the remaining potential is concentrated in South Africa, Namibia, Botswana, and Angola (Fig 1a). Wind potential is also widely distributed across countries even with socio-environmental protections, although in Angola, Mozambique, and the Democratic Republic of Congo (DRC) wind potential is limited to smaller areas (Fig 1b) due to either low wind speeds and/or extensive forest cover.”
R3–6	P4 Fig 1: P5 Section 2.2: It should be made clear here which demand growth is assumed over the modelling period, and to which per-capita electricity consumption by 2050 this corresponds.	We assumed the base electricity demand forecast scenario from the SAPP plan, which assumes a doubling of electricity demand from 2020 to 2040 and an average electricity consumption of 1,600 kWh per capita across the region (as per page 15 of the Main Volume of the SAPP Pool Plan (2017)). We added the following sentence to Results subsection 2.2: “We assume an electricity demand forecast from the base scenario of the Southern African Power Pool Plan (33).” And we elaborate on this in the Methods section (addition in green): “Hourly time series of electricity demand are based on actual 2018 data linearly extrapolated across investment periods assuming growth rates from the base scenario of the SAPP Plan (33). This scenario assumes a doubling of electricity demand from 2020 to 2040 and assumes an average electricity consumption of 1600 kWh per capita by 2040.”
R3–7	P8 Fig 3: I would also show the buildout of transmission lines under the different scenarios in this figure. Do higher environmental constraints for hydro and VRE mean more or less need for cross-regional exchanges?	Thank you for this suggestion, an idea that we coincidentally had actually considered when we prepared Figure 3 for the initial submission. We generated the transmission figure, but discovered that the selected new transmission capacity does not change significantly/notably between the scenarios and thus we decided to put this transmission figure in the Supplementary Information (SI). See Figure S6. Transmission Flows in the original SI. For ease, we copied and pasted the figure below (See SI Fig. S1 for country name labels):

		New Transmission Capacity Legal All Exclusions All Exclusions - No new Hydro Capacity (MW) — 0 — 2000 — 4000 — 6000 As you can see, most line capacities are similar between scenarios, with the exception of slightly more capacity on the corridor joining Tanzania, Zambia, and Namibia and slightly less capacity between Botswana and South Africa with more land use protections and/or less hydropower selected. Rather than showing this figure in the main body of the text, we have instead added the following text to the end of subsection 2.2 in the Results section: “These differences in geographic distribution of hydropower projects between scenarios resulted in only modest differences in additional international transmission capacity requirements (SI Fig. S6). There is slightly more transmission capacity on the corridor joining Tanzania, Zambia, and Namibia and slightly less transmission capacity between Botswana and South Africa with more siting protections and/or less hydropower selected. This was likely due to more proportional compensating increases in wind, solar PV, and/or natural gas capacity additions domestically.”
R3–8	P14 Line 38: “adjusting the coarse spatial resolution data to match the annual average wind speeds from the finer spatial resolution Global Wind Atlas (GWA) data”—can the authors expand on this? Is it similar to the method described in Sterl et al. 2022 https://www.nature.com/articles/s41597-022-01786-5? (In fact, I would generally recommend to have a look at that paper which	Thank you for pointing us to the paper by Sterl et al. 2022. We created our wind and solar zones and capacity factor data sets as part of an earlier analysis — Chowdhury et al. published in August 2022 in Joule before Sterl et al’s publication in October 2022. See https://www.sciencedirect.com/science/article/pii/S254243512200304X#mmc1. The Chowdhury et al. (2022) method is indeed generally similar to Sterl et al. (2022). While our linear bias correction method is similar for solar GHI, Sterl et al. (2022) implemented a more detailed bias-correction method for wind speeds in order to maintain the Weibull shape of the distribution. We have now added (new text in green) that we “apply a linear bias correction to the coarse spatial resolution data to match the annual average wind speeds from the finer spatial resolution Global Wind

	assesses model-ready datasets for solar PV and wind power across Africa, also taking into account land use constraints and protected areas: https://www.nature.com/articles/s41597-022-01786-5.)	Atlas (GWA) data, following the approach detailed in Chowdhury et al. (2022) (26). This approach is comparable to Sterl et al. (2022) (55)."
R3-9	P15 Line 26: "The water release for each reservoir was determined by dam-specific rule curves accounting for the reservoir water level, inflow, storage capacity, and downstream water requirements"—If this is the case, then how did the capacity expansion model decide on the hydropower dispatch at sub-daily level? It sounds as if the release was pre-ordained before model entry, but I am not sure if this was really the case.	You are correct that the water release for each reservoir, and thus the energy budget, is determined at daily resolution by using a hydrological and water management model (VIC-Res), which was run prior to the capacity-expansion model. VIC-Res provided a 20-year-long (1997-2016) time series of the daily energy budget, from which we estimated the average daily energy budget for each calendar month. The daily energy budgets were used in the capacity expansion model (GridPath-SAPP) which can then determine the hourly dispatch for each reservoir within a maximum limit fixed by the rated capacity of the hydropower plant and a minimum limit, assumed to be 30% of the average daily energy budget, including environmental flow constraints. For candidate hydropower projects, this dispatch is non-zero if the GridPath-SAPP chooses to build the project when the model co-optimizes investment decision and dispatch. To clarify this sequencing and passing of assumptions between the two models, we added the following to the Methods section: "Using these average daily energy budgets estimated by VIC-Res, GridPath-SAPP determines the hourly dispatch for each reservoir within a maximum limit (determined by the rated capacity) and a minimum limit (assumed to be 30% of the average daily energy budget), ensuring maximum generation for each day does not exceed the daily energy budget."

REVIEWERS' COMMENTS

Reviewer #3 (Remarks to the Author):

The authors have sufficiently addressed my comments. I recommend this paper for publication.